# In silico design of multi-epitope vaccines against the hantaviruses by integrated structural vaccinology and molecular modeling approaches

Liaqat Ali[1]⊕*, Sobiah Rauf[1]⊕, Abbas Khan[2]⊕, Samreen Rasool[3], Rabail Zehra Raza[1], Fahad M. Alshabrmi[4], Taimoor Khan[5], Muhammad Suleman[6], Yasir Waheed[7], Anwar Mohammad[8], Abdelali Agouni[2]*

1 Department of Biological Sciences, National University of Medical Sciences (NUMS), Rawalpindi, Pakistan, 2 Department of Pharmaceutical Sciences, College of Pharmacy, QU Health, Qatar University, Doha, Qatar, 3 Department of Biochemistry, Government College University, Lahore, Pakistan, 4 Department of Medical Laboratories, College of Applied Medical Sciences, Qassim University, Buraydah, Saudi Arabia, 5 Department of Radiation Oncology, University of California, San Francisco, United States of America, 6 Centre for Biotechnology and Microbiology, University of Swat, Charbagh, Swat, Khyber Pakhtunkhwa, Pakistan, 7 Gilbert and Rose-Marie Chagoury School of Medicine, Lebanese American University, Byblos, Lebanon, 8 Department of Biochemistry and Molecular Biology, Dasman Diabetes Institute, Dasman, Kuwait

⊕ These authors contributed equally to this work.
* liaqat.ali@numspak.edu.pk (LA); Aagouni@qu.edu.qa (AA)

**Data Availability Statement:** All relevant data are within the article and its Supporting Information files.

## Abstract

Hantaviruses are single-stranded RNA viruses belonging to the family *Bunyaviridae* that causes hantavirus cardiopulmonary syndrome (HCPS) and hemorrhagic fever with renal syndrome (HFRS) worldwide. Currently, there is no effective vaccination or therapy available for the treatment of hantavirus, hence there is a dire need for research to formulate therapeutics for the disease. Computational vaccine designing is currently a highly accurate, time and cost-effective approach for designing effective vaccines against different diseases. In the current study, we shortlisted highly antigenic proteins i.e., envelope, and nucleoprotein from the proteome of hantavirus and subjected to the selection of highly antigenic epitopes to design of next-generation multi-epitope vaccine constructs. A highly antigenic and stable adjuvant was attached to the immune epitopes (T-cell, B-cell, and HTL) to design Env-Vac, NP-Vac, and Com-Vac constructs, which exhibit stronger antigenic, non-allergenic, and favorable physiochemical properties. Moreover, the 3D structures were predicted and docking analysis revealed robust interactions with the human Toll-like receptor 3 (TLR3) to initiate the immune cascade. The total free energy calculated for Env-Vac, NP-Vac, and Com-Vac was -50.02 kcal/mol, -24.13 kcal/mol, and -62.30 kcal/mol, respectively. *In silico* cloning, results demonstrated a CAI value for the Env-Vac, NP-Vac, and Com-Vac of 0.957, 0.954, and 0.956, respectively, while their corresponding GC contents were 65.1%, 64.0%, and 63.6%. In addition, the immune simulation results from three doses of shots released significant levels of IgG, IgM, interleukins, and cytokines, as well as antigen clearance over time, after receiving the vaccine and two booster doses. Our vaccines

**Funding:** This work was supported by Qatar National Research Fund [grant No. NPRP14S-0406-210150] and Qatar University grant No. QUPD-CPH-23/24-592. The statements made herein are solely the responsibility of the authors. Open Access funding is provided by the Qatar National Library. Thank you to the authors for the collaborative work and their cooperation. We would extend our thanks to National University of Medical Sciences for providing us a platform to facilitate present study.

**Competing interests:** The authors have declared that no competing interests exist.

against Hantavirus were found to be highly immunogenic, inducing a robust immune response that demands experimental validation for clinical usage.

## Introduction

Hantaviruses are single-stranded tri-segmented enveloped RNA viruses that belong to the family Bunyaviridae. According to the International Committee for the Taxonomy of Viruses (ICTV), they form their own family *Hantaviridae* within the order *Bunyavirales* [1]. These viruses infect many species of rodents, shrews, moles, and bats. Infections in these reservoir hosts are almost asymptomatic, but some rodent-borne hantaviruses also infect humans causing either hemorrhagic fever with renal syndrome (HFRS) or hantavirus cardiopulmonary syndrome (HCPS) [2]. Hantaviruses pose an emerging global threat to public health causing a devastating effect on human lives, affecting more than 200,000 individuals worldwide annually [3]. Moreover, the number of cases is significantly increasing day by day in different parts of the world [4].

The genome of hantaviruses contains three segments according to their nucleotide sequence length i.e., small (S), medium (M), and large (L) [5]. The envelope membrane is composed of a bilayer of exterior lipids lined with spikes of viral proteins, appears as heterodimers of Gn and Gc glycoproteins, and shows a significant binding affinity with oligomers [6]. The variation and genetic alterations in M and S segments can disturb the virulence and antigenicity of the virus [7]. In general, these viruses are classified into three different classes based on their associated reservoir host. The first class is of old-world viruses such as Seoul virus (SEOV), Hantaan virus (HNTV), and Dobrava–Belgrade virus (DOBV) responsible for HFRS. These are hosted by Murinae rodents predominantly found in Europe and Asia. The second class consists of New World viruses such as New York-1 virus (NY-1V), Sin Nombre virus (SNV), and Andes virus (ANDV), causing hantavirus pulmonary syndrome (HPS). These viruses are transmitted via Sigmon-dontinae subfamily members mostly found in the US. The third class comprises both Old and New World hantaviruses such as Prospect Hill Virus (PHV), Puumala virus (PUUV), and Tula virus (TULV) harbored by Arvicolinae rodents [8, 9]. Recently, many bioinformatics studies have revealed that the progression of hantaviruses is a consequence of the coevolution of viruses from bats, shrews, and moles to rodents that is responsible for the emergence of virulent hantaviruses capable of infecting humans [9–11].

Currently, there are no approved vaccines or antiviral therapies available for the treatment of these hantavirus diseases. Hantavax is a commercialized, formalin-inactivated vaccine for HFRS, prepared against HTNV growth inside the brains of mice [12]. Initially, Hantavax was found to elicit a strong immune response; however, it failed to establish a statistically significant decrease in vaccine HFRS disease intensity [13]. An optimal multi-epitope vaccine construct should have a sequence of overlapping epitopes, with each antigenic peptide fragment eliciting either a cellular or a humoral immune response against the tumor or virus of interest [14]. Therefore, our present study aims to focus on developing a multi-epitope vaccine against hantavirus, in order to activate an immunological response. Therefore, we selected the non-overlapping, non-allergenic, and highly antigenic epitopes. The sequences of the designed multi-epitope vaccine constructs of Env-Vac, NP-Vac, and Com-Vac were modeled and after evaluation docked with TLR3 for interaction analysis. Evaluations were then completed to confirm vaccine stability and effectiveness. Finally, the antigenicity, allergenicity, physiochemical properties molecular docking, *in silico* cloning, and immune simulation, further validated

the designed constructs that can be used to check the clinical safety and efficacy of the candidate vaccine for clinical usage against hantaviruses.

## Materials and methods

### Sequence retrieval and antigenicity check

The proteome of Hantavirus was retrieved from Hantaviruses DB database using proteome ID UP000204348 [15]. Envelope polyprotein (UniProt ID: Q9E006) and Nucleoprotein (UniProt ID: O36307) were the probable antigens and the results were matched by server VaxiJen (http://www.ddg-pharmfac.net/vaxijen/VaxiJen/VaxiJen.html) using default threshold of 0.4 [16]. Both of the antigenic proteins were subjected for allergenicity tests using AllerTop v2.0 (https://www.ddg-pharmfac.net/AllerTOP/) [17].

### Selection of immune epitopes

NetCTL1.2 (https://services.healthtech.dtu.dk/service.php?NetCTL-1.2) with a default value of 0.75 was employed to predict cytotoxic T lymphocytes (CTL) epitopes [18, 19]. Helper T cell epitopes were predicted by using the IEDB (The Immune Epitope Database) server (http://tools.iedb.org/mhcii/), which uses 7 sets of alleles for human leucocyte antigens (HLAs) [20]. ABCPred server (https://webs.iiitd.edu.in/raghava/abcpred/ABC_submission.html) with a cut-off score of 0.8 was used for the prediction of B cell epitopes (a key in generating the protective host antibody responses) [21]. Each epitope was subjected to antigenic analysis to further screen and shortlist predicted epitopes and the highly antigenic epitope was selected for vaccine construction. ElliPro (http://tools.iedb.org/ellipro/) was used for the calculation of conformational B cell epitopes in the vaccine structure [22].

### Construction of multi-epitope vaccine

Different linkers were used to combine the favorable properties of all selected CTL, HTL, and B cell epitopes in a single vaccine. The final vaccine constructs consisted of adjuvant, CTL, HTL, and B cell, joined together by AAY, GPGPG, and KK linkers, respectively. At the N-terminus of the vaccine, an adjuvant was used to stabilize and enhance the immune responses [23, 24]. Antigenicity and allergenicity tests were again done on the final sequence of constructed vaccines using VaxiJen and AllerTop v2.0 (https://www.ddg-pharmfac.net/AllerTOP/), respectively.

### Prediction of physicochemical properties and secondary structure analysis

An online tool called ProtParam (https://web.expasy.org/protparam/) [25, 26] was employed to collect information about physiochemical properties such as molecular weight, theoretical PI, GRAVY, aliphatic index, instability index, grand average of hydropathicity, and half-life of the final vaccine construct. The secondary structure of each construct was predicted by using PDBsum. The solubility for each construct was estimated by using the online server Protein-sol (https://protein-sol.manchester.ac.uk/) which calculates the solubility of a protein by comparing it with the experimental dataset [27].

### Structure modeling and evaluation

Robetta (http://robetta.bakerlab.org/) which uses the comparative modeling approaches to predict structure, was used for 3D structure modeling [28]. From the predicted 5 models, one was selected based on the highest score. Validation and quality check of selected models was

done using RAMPAGE (http://mordred.bioc.cam.ac.uk/rapper/rampage.php) and ProSA-web (https://prosa.services.came.sbg.ac.at/prosa.php) [29, 30].

## Molecular docking and MM-GBSA analysis

The binding interaction of constructed vaccines with the human toll-like receptor-3 (TLR-3) was analyzed using the HDOCK server (http://hdock.phys.hust.edu.cn/) [31]. The HDOCK server employs a hybrid algorithm that integrates both template-based and template-free methods for the automated prediction of protein-protein interactions. This server sets itself apart from other similar services due to its capacity to accommodate amino acid sequences as input files. In order to evaluate the binding free energies of the vaccine-TLR complex, the MM/GBSA approach was utilized, which was facilitated by accessing the HawDock online server (http://cadd.zju.edu.cn/hawkdock/ [32].

## In silico cloning and codon optimizations

To improve protein expression, a computer-based genetic technique called in silico codon optimization is commonly used. In this regard, we utilized the Java codon adaptation tool (JCat) available at http://www.jcat.de/ to transform the amino acid sequence of the vaccine into a DNA sequence, enabling its cloning in the *Escherichia coli (E. coli)* K12 expression system. By examining the GC content and codon adaptation index (CAI) values obtained through JCat, we could assess the level of expression achievable in the *E. coli* expression system. To complete the process, we employed SnapGene software for cloning the developed vaccines into the pET-28a (+) expression vector [33–36].

## Immune simulation

We assessed the human immune system's response to the constructed vaccines, including envelope, nucleoprotein, and proteome-wide vaccines, by employing an online immune simulation server known as C-ImmSim (https://kraken.iac.rm.cnr.it/C-IMMSIM/index.php?page=1) [37]. Three injections were administered at different time intervals (0, 17, and 56th day) and set other parameters as default. This server enabled us to evaluate the antigenicity of the vaccines and their ability to elicit immune responses upon injection. Through this server, we determined the level of helper T-cell 1 (Th1) and helper T-cell 2 (Th2) immune cells generated in response to the vaccines, as well as the production of interferon, cytokines, and antibodies, among other immunological responses, against the administered vaccines. The methodological flow of the work is given in **Fig 1**.

## Results

In order to remove and neutralize invading pathogens, it is crucial to rely on the acquired or adaptive immune responses [29]. When the immune system encounters the same pathogens again, it generates memory immune cells that can identify the targeted pathogen and remember the antibodies that can fight against that particular invader. This information forms the basis for creating vaccines [30]. Computational modeling and combined immune informatics approaches were implemented to make a multi-epitope vaccine for the development of immunity against the hantavirus. The total proteome of hantavirus contains three different proteins Envelope polyprotein, Replicase, and Nucleoprotein. Two proteins (Envelope and Nucleoprotein) are probable antigens which was again confirmed using VaxiJen 2.0 with values 0.5265 and 0.5770, respectively. Both are non-allergens as per the results of the allergenicity test. Putative B cell, cytotoxic T cell, and helper T cell epitopes were screened from these antigenic proteins.

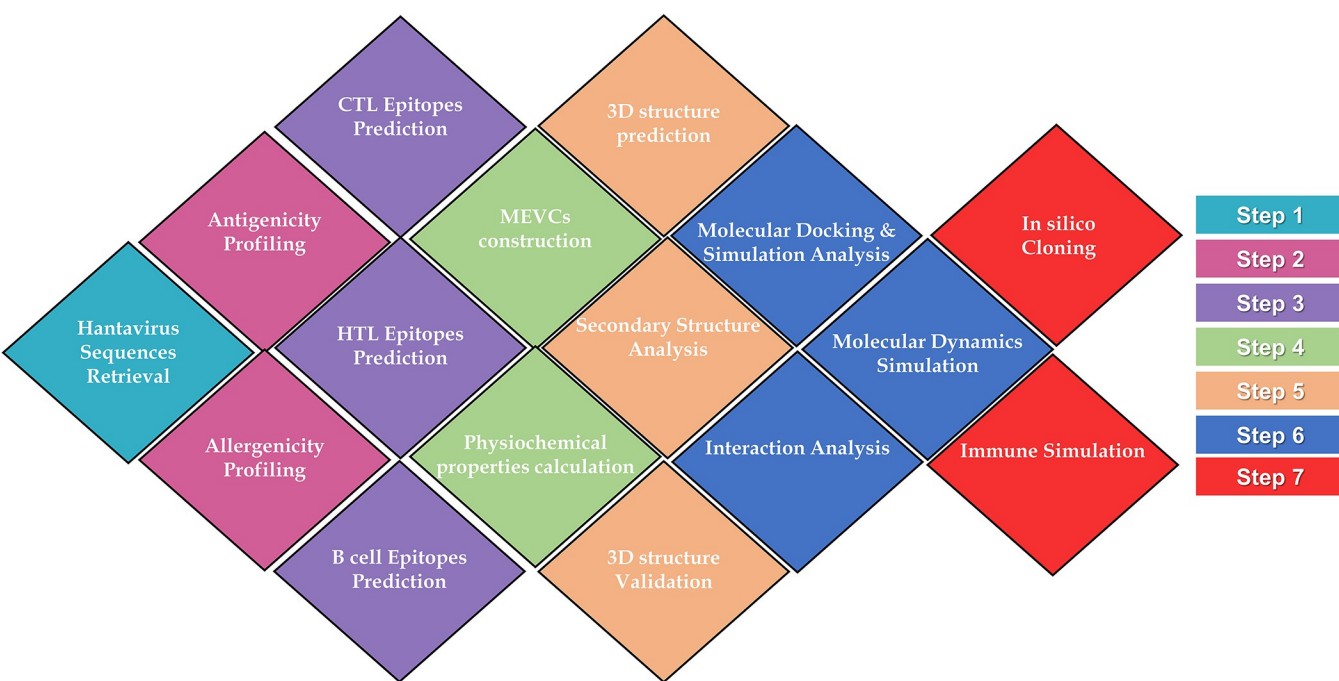

**Fig 1. Systematic workflow of the study design.** Each colour represents the specific step which can be recognized by the colour scheme given in the right panel.

### Immunogenic CTL epitopes prediction

Selected proteins (Envelope, Nucleoprotein, and proteome-wide) were screened to predict the immunogenic CTL epitopes. In Envelope, and Nucleoprotein, a total of 1130 and 420 CTL peptides were predicted, of which only 32 and 7 were predicted to be MHC ligands, respectively. On the basis of high antigenicity, 3 epitopes were selected for each envelope (QTHCQPTVY, ILEKVKIEY, HTVGLGQGY) and nucleoprotein (LKEKSSLRY, FMSTNKMYF, KLKKKSAFY). All these 6 highest antigenicity epitopes were selected from both of proteins for the proteome-wide vaccine construct. Final CTL epitopes were based on values of antigenicity, C-terminal cleavage, and TAP score as summarized in **Table 1**.

### Immunogenic HTL epitopes prediction

The next step was the prediction of helper T cell epitopes to trigger the immune response. For each of the proteins, 6 of the lowest percentile HTL epitopes were selected from the list of predicted epitopes by the online server. 4 epitopes which have highest antigenic score were further shortlisted for Envelope (VVLVVILILSIIMFS-HLA-DRB4*01:01, VVVLVVILILSIIMF-HLA-DRB4*01:01, LVVILILSIIMFSVL- HLA-DRB4*01:01, VLVVILILSIIMFSV-HLA-DRB4*01:01) and Nucleoprotein (MGIQLDQKIIILYML- HLA-DRB1*03:01, QSMGIQLDQKIIILY- HLA-DRB1*03:01, SMGIQLDQKIIILYM- HLA-DRB1*03:01, TQSMGIQLDQKIIIL0-HLA-DRB1*03:01). From both Envelope and Nucleoprotein 6 HTL epitopes with highest score were selected for the construction of whole proteome vaccine (**Table 2**).

### Immunogenic B-cell epitopes prediction

Epitopes with a length of 20-mer and a score 0.9 and above were selected. After the antigenicity check, two epitopes with the highest antigenic score were finalized. And for the construction

**Table 1. Summary of selected CTL epitopes for envelope and nucleoprotein of Hantavirus.**

| Residue no | Peptide sequence | MHC Binding affinity | Rescale binding affinity | C-terminal cleavage affinity | Transport affinity | Prediction score | MHC-I Binding | Antigenicity score (0.4) |
|---|---|---|---|---|---|---|---|---|
| | | | | Envelope | | | | |
| 139 | QTHCQPTVY | 0.50 | 2.13 | 0.94 | 3.08 | 2.42 | Yes | 1.04 |
| 531 | ILEKVKIEY | 0.31 | 1.33 | 0.97 | 2.71 | 1.61 | Yes | 0.92 |
| 32 | HTVGLGQGY | 0.29 | 1.26 | 0.80 | 2.74 | 1.52 | Yes | 0.69 |
| | | | | Nucleoprotein | | | | |
| 90 | LKEKSSLRY | 0.25 | 1.10 | 0.92 | 2.81 | 1.37 | Yes | 0.70 |
| 257 | FMSTNKMYF | 0.16 | 0.69 | 0.77 | 2.71 | 0.94 | Yes | 0.56 |
| 353 | KLKKKSAFY | 0.11 | 0.49 | 0.96 | 2.88 | 0.77 | Yes | 1.27 |
| | | | | Proteome | | | | |
| 139 | QTHCQPTVY | 0.50 | 2.13 | 0.94 | 3.08 | 2.42 | Yes | 1.04 |
| 531 | ILEKVKIEY | 0.31 | 1.33 | 0.97 | 2.71 | 1.61 | Yes | 0.92 |
| 32 | HTVGLGQGY | 0.29 | 1.26 | 0.80 | 2.74 | 1.52 | Yes | 0.69 |
| 90 | LKEKSSLRY | 0.25 | 1.10 | 0.92 | 2.81 | 1.37 | Yes | 0.70 |
| 257 | FMSTNKMYF | 0.16 | 0.69 | 0.77 | 2.71 | 0.94 | Yes | 0.56 |
| 353 | KLKKKSAFY | 0.11 | 0.49 | 0.96 | 2.88 | 0.77 | Yes | 1.27 |

of a proteome-wide vaccine, all 4 epitopes from both proteins were considered. B cell epitopes are summarized in **Table 3**.

## Construction of multi-epitope vaccine construct

The adjuvant (nontoxic human beta defensin-2) is attached to the N terminal end of the vaccine to enhance the immunogenicity of epitopes. From the selected CTL, HTL, and B cell epitopes (**Tables 1, 2, and 3**), a full-length multi-epitope vaccine construct was made using different linkers, such as EAAAK, GPGPG, AAY, and KK for joining. Adjuvant was joined to

**Table 2. Summary of Helper T-cell epitopes selected for envelope and nucleoprotein of Hantavirus using the IEDB MHC-II module.**

| Sr. No. | Allele | Method | Peptides | Percentile Rank | Antigenicity score (0.4) |
|---|---|---|---|---|---|
| | | | Envelope | | |
| 1 | HLA-DRB4*01:01 | Consensus | VVLVVILILSIIMFS | 0.02 | 0.59 |
| 2 | HLA-DRB4*01:01 | Consensus | VVVLVVILILSIIMF | 0.02 | 0.50 |
| 3 | HLA-DRB4*01:01 | Consensus | LVVILILSIIMFSVL | 0.08 | 0.53 |
| 4 | HLA-DRB4*01:01 | Consensus | VLVVILILSIIMFSV | 0.09 | 0.54 |
| | | | Nucleoprotein | | |
| 1 | HLA-DRB1*03:01 | Consensus | MGIQLDQKIIILYML | 0.05 | 0.76 |
| 2 | HLA-DRB1*03:01 | Consensus | QSMGIQLDQKIIILY | 0.05 | 0.76 |
| 3 | HLA-DRB1*03:01 | Consensus | SMGIQLDQKIIILYM | 0.05 | 0.84 |
| 4 | HLA-DRB1*03:01 | Consensus | TQSMGIQLDQKIIIL | 0.06 | 0.73 |
| | | | Proteome | | |
| 1 | HLA-DRB4*01:01 | Consensus | VVLVVILILSIIMFS | 0.02 | 0.59 |
| 2 | HLA-DRB4*01:01 | Consensus | LVVILILSIIMFSVL | 0.08 | 0.53 |
| 3 | HLA-DRB4*01:01 | Consensus | VLVVILILSIIMFSV | 0.09 | 0.54 |
| 4 | HLA-DRB1*03:01 | Consensus | MGIQLDQKIIILYML | 0.05 | 0.76 |
| 5 | HLA-DRB1*03:01 | Consensus | QSMGIQLDQKIIILY | 0.05 | 0.76 |
| 6 | HLA-DRB1*03:01 | Consensus | SMGIQLDQKIIILYM | 0.05 | 0.84 |

**Table 3. Results of ABCPred: Summary of linear B cell epitopes.**

| Sr. No. | Epitope | Position | Score | Antigenicity score (0.4) |
|---|---|---|---|---|
| Envelope | | | | |
| 1 | LIILKCLRVLTFSCSHYTNE | 506 | 0.96 | 0.62 |
| 2 | KTDLELDFSLPSSSSYSYRR | 673 | 0.90 | 0.95 |
| Nucleoprotein | | | | |
| 1 | KDAEKAVEVDPDDVNKSTLQ | 26 | 0.91 | 0.61 |
| 2 | FPAQVKARNIISPVMGVIGF | 206 | 0.90 | 0.99 |
| Proteome | | | | |
| 1 | LIILKCLRVLTFSCSHYTNE | 506 | 0.96 | 0.62 |
| 2 | KTDLELDFSLPSSSSYSYRR | 673 | 0.90 | 0.95 |
| 3 | KDAEKAVEVDPDDVNKSTLQ | 26 | 0.91 | 0.61 |
| 4 | FPAQVKARNIISPVMGVIGF | 206 | 0.90 | 0.99 |

the CTL epitopes using EAAAK linker and all the selected CTL epitopes were joined using AAY linkers. To join all of the selected HTL and B cell epitopes, GPGPG and KK linkers were used, respectively as shown in **Fig 2**. The topological constructs of each vaccine candidate are given in **Fig 2A–2C**, where an adjuvant is attached to three CTL, four HTL, and two B cell epitopes in the envelope-based vaccine construct. The selection of specific epitopes is based on the predicted scores by the machine learning algorithms, and the calculated antigenicity and allergenicity profiles. Moreover, in the vaccine construct from nucleoprotein, the same adjuvant is attached to three CTL, five HTL, and two B cell epitopes. Similarly, the combined

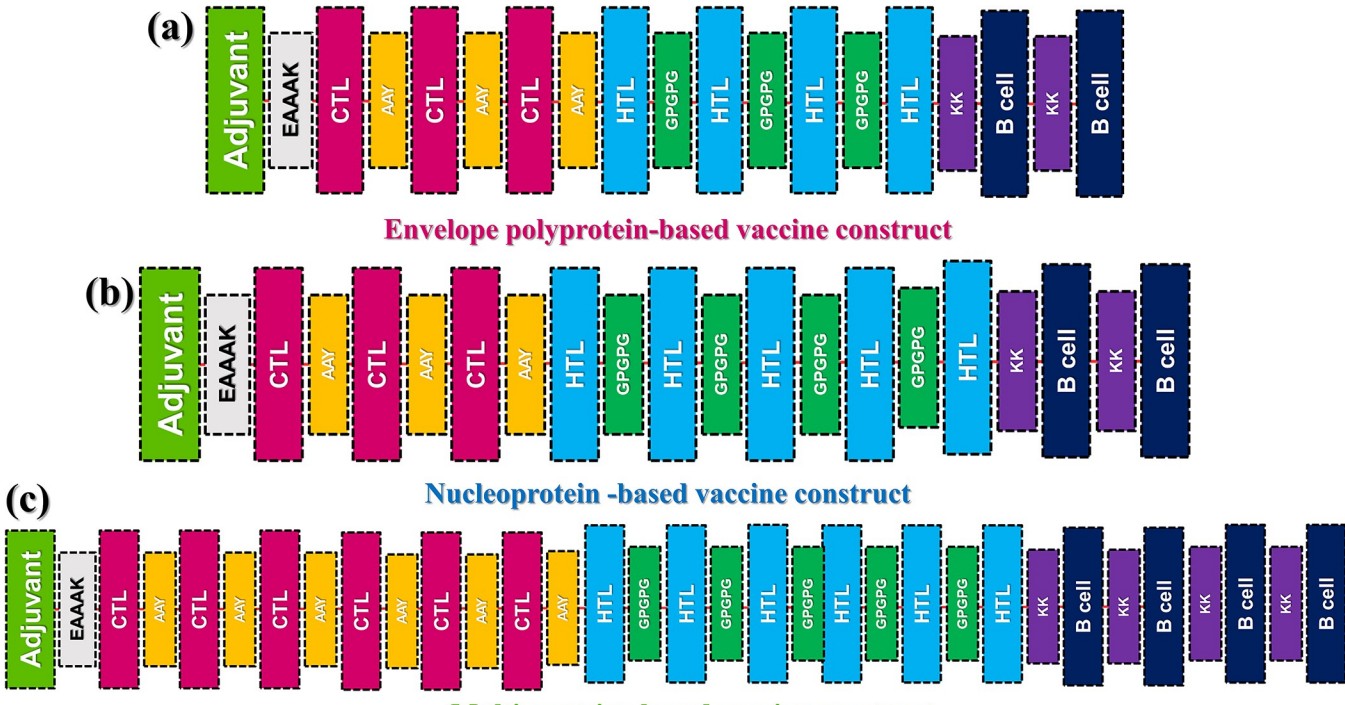

**Fig 2. Topological representation of the vaccine constructs.** (**a**) represent the topological architecture of the Env-Vac construct from envelope polyprotein, (**b**) represent the topological arrangement of the predicted epitopes from nucleoprotein NP-Vac construct while (**c**) represents the topological organization of the epitopes from different proteins named as Com-Vac.

vaccine construct is made up of five CTL, five HTL, and four B cell epitopes. All vaccine constructs are non-allergens and probable antigens with values of 0.61, 0.64, and 0.62, respectively. This indicates their capability to provoke immune responses with no allergenic reaction and can be considered for further analysis.

## Structure modeling of multi-epitope vaccine constructs and validation

The 3D structures of the designed constructs are given in **Fig 3A–3C**. To highlight epitopes and linkers different colors were used and different distinct representation styles were also used in structure models. The specific epitopes are presented in the same colour in each construct and the linkers are also coloured differently for clarity. In each epitope, the adjuvant given in green colour represents the adjuvant joined by EAAAK linker given in grey colour while the CTL epitopes are shown in red colour joined by AAY linkers in black colour. Moreover, the cyan colour represents HTL epitopes in each construct while the B cell epitopes are given in black.

Structural validation of the modeled vaccine constructs was carried out to see the structural and functional relevance of the folding. We used multiple parameters to validate the 3D structures of constructed models. Ramachandran plot shows the maximum residues of all models in the favored region (**Fig 4A–4C**). Highly preferred observations with values 180 (97.826%), 168 (93.333%), and 273 (95.455%) were observed for envelope, nucleoprotein, and proteome-wide, respectively. The Z-scores of the input models were found to be -2.98, -6.8, -5.96. for the envelope, nucleoprotein, and proteome-wide, respectively (**Fig 4D–4F**) which were normal. Therefore, the selected models can be subjected to next step of molecular docking.

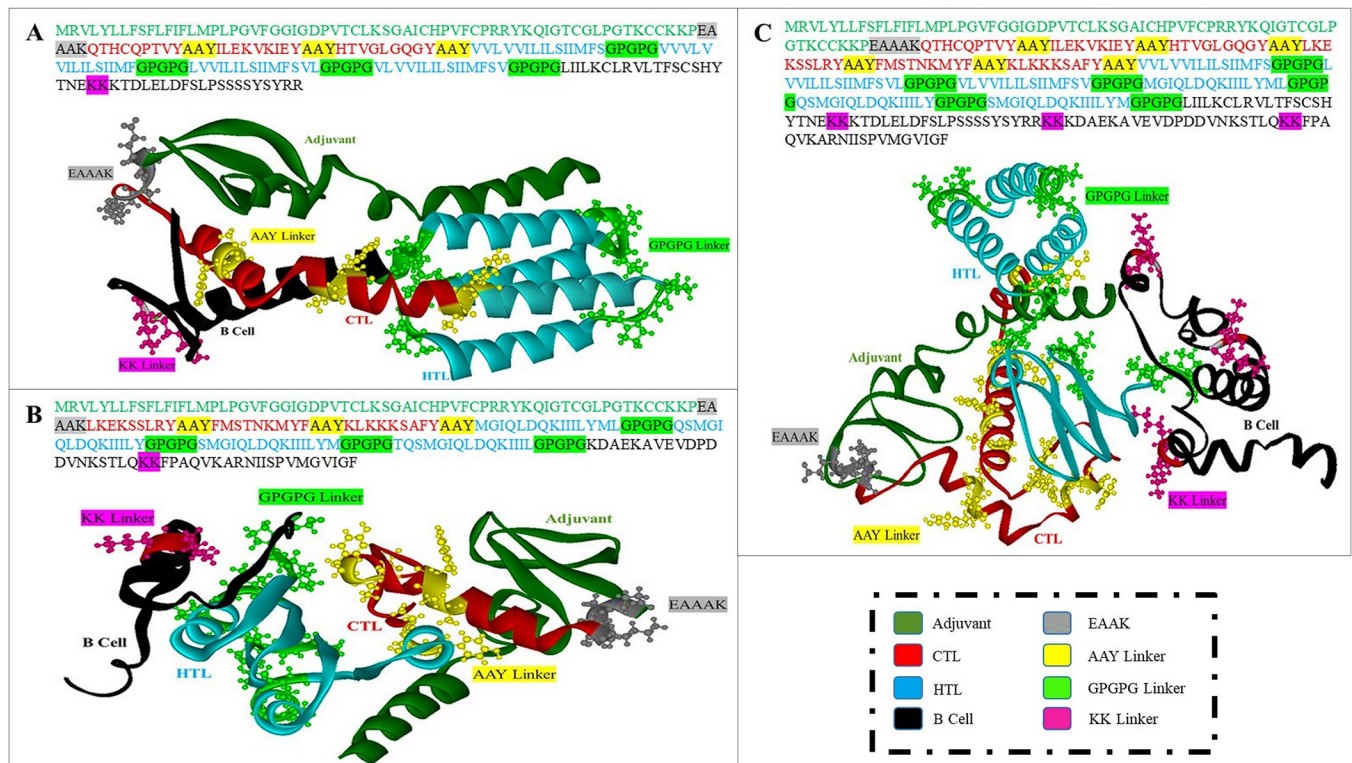

**Fig 3. Structural modeling and epitopes mapping for each construct.** (**A**) represent the 3D structure of envelope protein-based MEVC where different colour represent different epitopes. (**B**) represent the 3D structure of envelope protein-based MEVC where different colour represent different epitopes. (**C**) represent the 3D structure of envelope protein-based MEVC where different colour represent different epitopes.

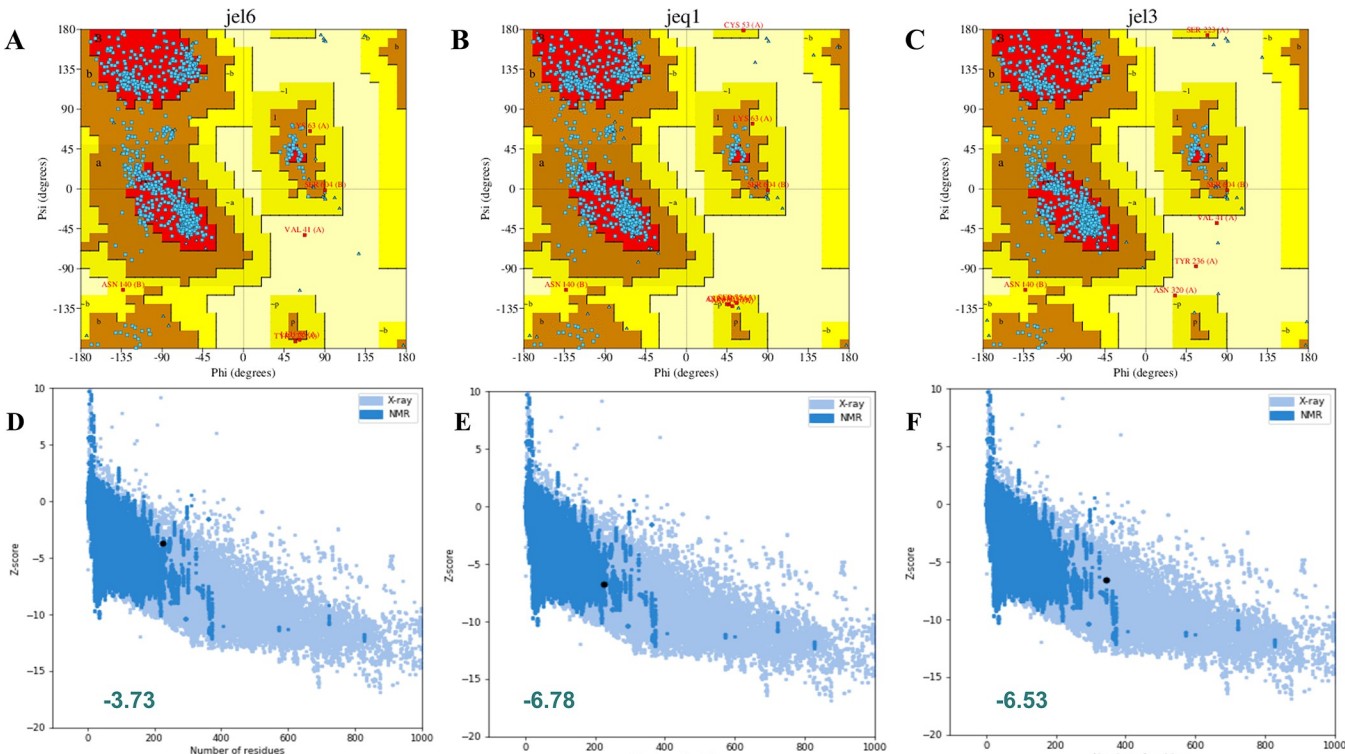

**Fig 4. Structural validation of the modeled vaccine constructs. (A-C)** show the Ramachandran plots for all the designed vaccine constructs. **(D-E)** show the ProSA-web results for all the designed vaccine constructs.

### Analysis of physicochemical properties and secondary structure analysis

The identified molecular weights and pI, for the envelope, nucleoprotein, and proteome-wide were 24.41, 24.68 37.85 kDa, and 9.23, 9.57, 9.51 respectively. pI values indicate that our candidate protein is acidic in nature. Half-life in *E. coli* was found to be > 10 for all three constructs. All vaccine constructs were determined to be stable, with an instability index of 42.62, 44.55, and 44.81 for envelope, nucleoprotein, and proteome-wide, respectively. The GRAVY and aliphatic index were found to be 0.902 and 127.96 for envelope, 0.150 and 94.96 for nucleoprotein, 0.410 and 109.02 for proteome-wide construct, respectively. The physiochemical properties of the vaccine constructs have been summarized in **Table 4**. Secondary structure elements, including alpha helices and beta sheets, are essential in determining a protein's overall structure, stability, and function. By minimizing the exposure of hydrophobic amino acids to solvent, both alpha helices and beta sheets contribute to a protein's overall stability, while also providing structural support that maintains its three-dimensional shape. Moreover, the specific arrangement of these secondary structures is critical to a protein's function. For instance, beta sheets in enzymes often form the active site where substrate binding and

**Table 4. The physiochemical properties of constructed vaccines.**

| Vaccine construct | Molecular weight (kDa) | Theoretical PI | Half-life in *E. Coli* | Instability index | Aliphatic index | GRAVY |
|---|---|---|---|---|---|---|
| **Env-Vac** | 24418.68 | 9.23 | >10 hours | 42.62 | 127.96 | 0.902 |
| **NP-Vac** | 24681.58 | 9.57 | >10 hours | 44.55 | 94.96 | 0.150 |
| **Com-Vac** | 37852.45 | 9.51 | >10 hours | 44.81 | 109.02 | 0.410 |

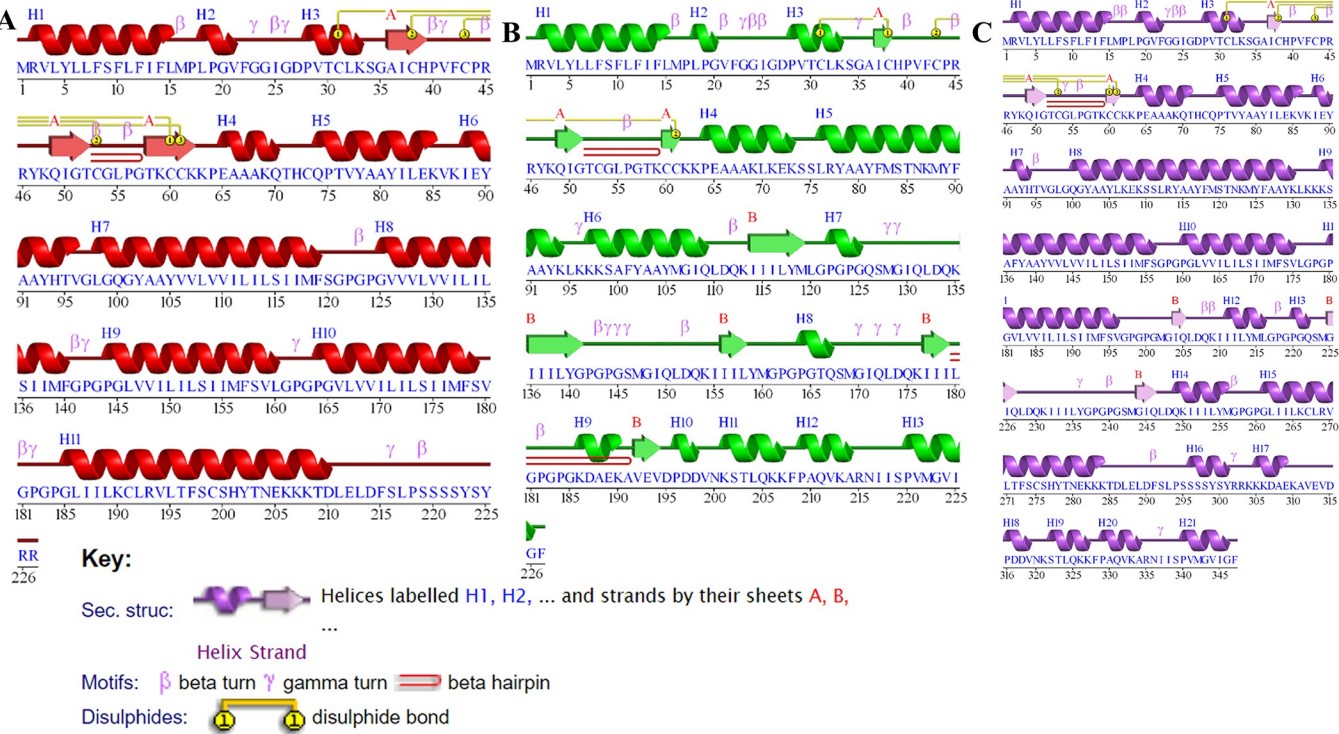

**Fig 5. Secondary structural elements distribution in each vaccine construct.** (**A**) show the secondary structure for Env-Vac, (**B**) show the SS for NP-Vac while (**C**) show the SS for Com-Vac.

chemical reactions occur, whereas alpha helices contribute to protein function by forming recognition sites for other proteins or molecules. Any changes in a protein's secondary structure, such as mutations or denaturation, can significantly impact its stability and function. For instance, mutations that modify crucial alpha helices or beta sheets can impede protein folding and disrupt its interactions with other molecules. Denaturation, resulting from extreme conditions such as high temperature or pH, can cause the protein to lose its native structure, leading to functional impairment. Understanding the essential role of SS elements in the functionality of a protein, we also estimated the SS elements in each constructed vaccine. It can be seen that each vaccine design has a larger number of alpha-helices and beta-sheets that are connected by the loops. This shows that our designed constructs have proper folding and thus consequently stability that will robustly trigger the immune response. The predicted secondary structures for each construct are given in **Fig 5A–5C**. Moreover, proteins with a scaled solubility value above 0.45 are predicted to be more soluble than the average soluble Escherichia coli protein from the experimental solubility dataset. For our proteins, i.e., for the Env-Vac the predicted solubility was 0.628, for the NP-Vac the predicted solubility was 0.641 while for the Com-Vac the solubility was calculated to be 0.544. The solubility assessment revealed that all the designed vaccine candidates are highly soluble thus ensuring that the vaccine components can be properly dissolved and administered within the body. The solubility graphs are given in **Fig 6A–6C**.

## Conformational B cell epitopes prediction

For the conformational B cell epitopes in each vaccine construct we used Ellipro server. For the Env-Vac four conformational B cell epitopes were reported while for the NP-Vac and

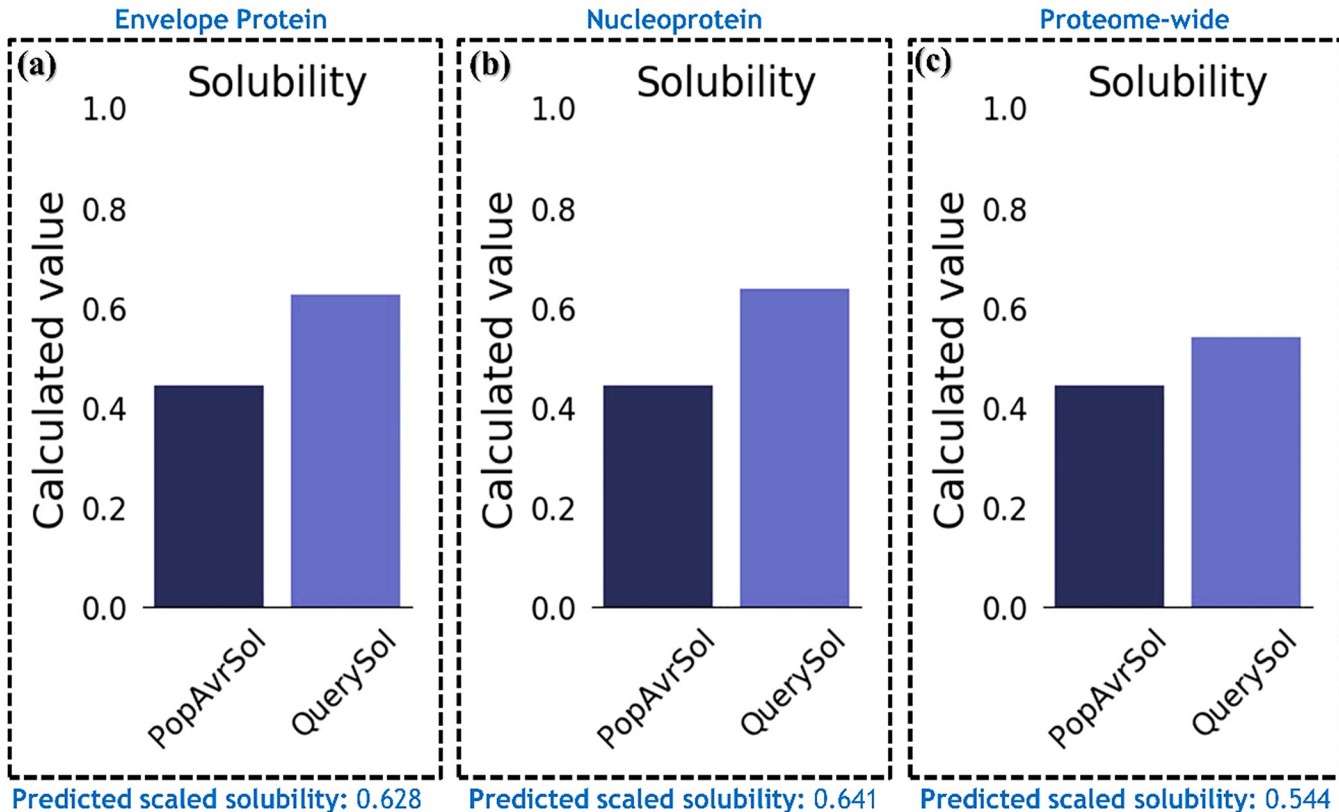

**Fig 6. Solubility analysis of each vaccine construct. (A)** show the solubility graph for Env-Vac, **(B)** show the solubility graph for NP-Vac and **(C)** show the solubility graph for Com-Vac.

Com-Vac five conformational B cell epitopes in each were calculated. The CBCEs in Env-Vac are given in **Fig 7A–7D**, the CBCEs in NP-Vac are given in **Fig 7E–7I** while the CBCEs in Com-Vac are given in **Fig 7J–7N**. The specific residues, scores and other information are given in **Table 5**.

## Molecular Docking analysis of the Vaccines-TLR3 complexes

For instance, the Env-Vac reported six hydrogen bonds, two salt bridges, and 238 non-bonded contacts. Among the hydrogen bonds, Cys199-Gln483, Cys199-Gln483, Thr203-Glu533, Lys207-Glu587, Ser224-Arg635 and Ser224-Arg635, while the two salt-bridges involve Lys207-Glu587, Asp210-Lys589 amino acids coupling. The interaction pattern for Env-Vac-TLR3 is given in **Fig 8A–8B**. On the other hand, the NP-Vac reported four hydrogen bonds, two salt bridges, and 259 non-bonded contacts. The hydrogen bonds involve Ser127-Tyr383, Ser127-Asn361, Asp197-Lys589, Asn200-Ser614 while the salt-bridges involve Lys97-Glu175, Asp197-Lys589 residues. The interaction pattern for NP-Vac-TLR3 is given in **Fig 8C–8D**. The proteome-wide complex reported eight hydrogen bonds, and one salt bridge while 211 non-bonded contacts between the TLR3 and Comb-Vac. Among the hydrogen bonds, Ile88-Trp273, Glu89-Lys272, Tyr93-Gly215, His94-Ala246, Ala139-Asn291, Phe173-Lys345, Ser174-His312 and Ser174-His312 while the only salt-bridge was established by Glu89-Lys272 residues. The interaction pattern for com-Vac-TLR3 is given in **Fig 8E–8F**. From the docking, it was observed that hydrogen bonding and hydrophobic interactions are dominant. It can be seen that residues involved in docking between TLR3-envelope and TLR3-nucleoprotein are

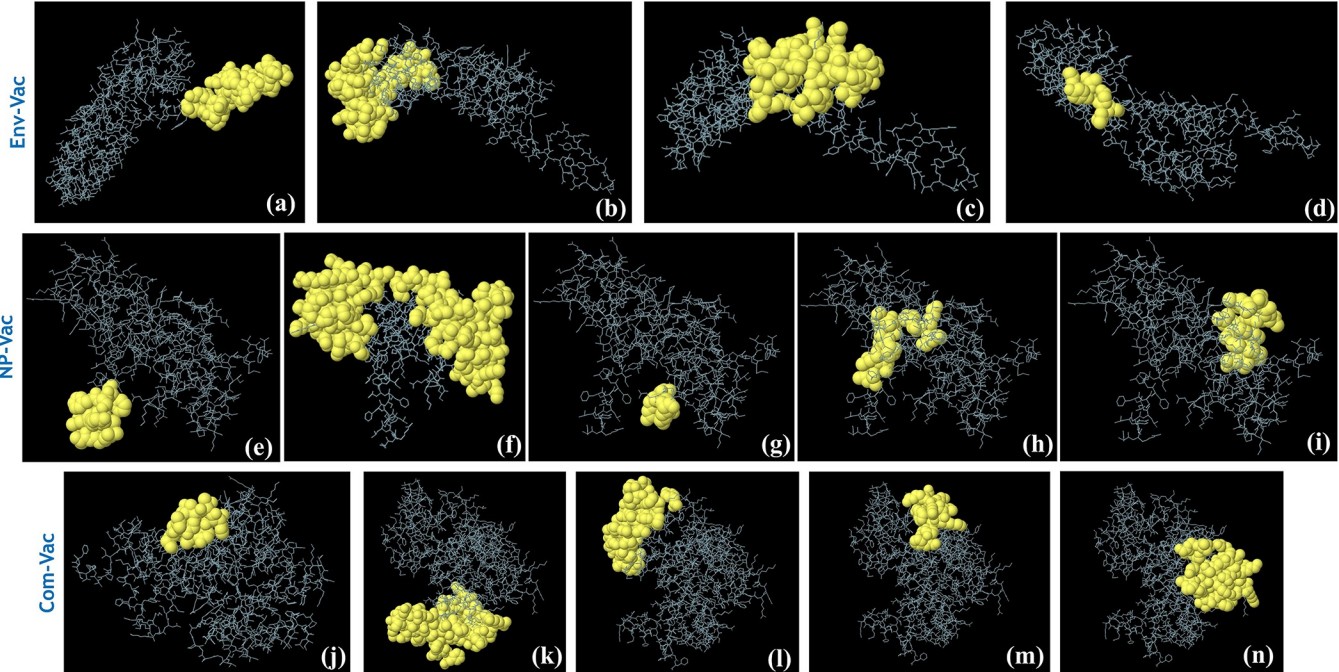

**Fig 7. The predicted conformational B cell epitopes for each vaccine construct. (a-d)** shows the CBCEs in Env-Vac, **(e-i)** shows the CBCEs in NP-Vac while (j-n) show the CBCEs in Com-Vac.

different from TLR3-proteome-wide which indicates a change in the active site for overall interaction of proteome-wide with TLR3. It was observed that there were 6, 4, and 7 hydrogen bonds for the envelope, nucleoprotein, and proteome-wide respectively.

### Estimation of the Gibbs binding free energy

Binding free energies were calculated for the Env-Vac, NP-Vac, and Com-Vac vaccine-TLR3 complex after MM-GBSA analysis. **Table 6** shows binding free energies for these docked complexes. The total energy calculated for the envelope, nucleoprotein, and proteome-wide is -50.02 kcal/mol, -24.13 kcal/mol, and -62.30 kcal/mol, respectively. Specific residues of ligands and receptors are involved in docking interaction. These ligand/receptor residues dock with the help of non-covalent interactions, for example, hydrogen bonding and hydrophobic interactions.

### In silico cloning of MESVs in the pET-28a (+) expression vector

To maximize the expression of the envelope, nucleoprotein, and proteome-wide vaccines in the *E. coli* K12 strain expression system, we performed codon optimization using the JCat tool. This tool calculates various parameters, including the CAI value and GC content [34]. The resulting CAI values for the envelope, nucleoprotein, and proteome-wide vaccines were 0.957, 0.954, and 0.956, respectively, while their corresponding GC contents were 65.1%, 64.0%, and 63.6%. These CAI values are within the desirable range for high expression, although the ideal GC content ranges from 30% to 70%. These results confirm that the constructed vaccines can be highly expressed in the *E. coli* expression system. Following optimization, we cloned the optimized sequences into the pET-28a (+) expression vector via Xho1 and EcoR1 restriction sites (**Fig 9**).

**Table 5. The predicted B cell conformational epitopes with their residues, size and scrores.**

| S. No | Residues | Size | Score |
|---|---|---|---|
| | **Envelope Protein** | | |
| 1 | A:T203, A:E205, A:K206, A:K207, A:K208, A:T209, A:D210, A:L211, A:E212, A:L213, A:D214, A:F215, A:S216, A:L217, A:P218, A:S219, A:S220, A:S221, A:S222, A:Y223, A:S224, A:Y225, A:R227 | 23 | 0.893 |
| 2 | A:M1, A:R2, A:V3, A:L4, A:Y5, A:L6, A:L7, A:F8, A:V110, A:I113, A:L114, A:I116, A:I117, A:M118, A:F119, A:S120, A:G121, A:P122, A:G123, A:P124, A:G125, A:V126, A:V127, A:V128, A:V130, A:V131, A:I134, A:L135, A:I137, A:I138, A:M139, A:G141, A:P142, A:I154, A:I155, A:M156, A:F157, A:S158, A:V159, A:L160, A:G161, A:P162, A:G163, A:P164, A:G165, A:V166, A:V168 | 47 | 0.722 |
| 3 | A:G24, A:I25, A:G26, A:S34, A:G35, A:V41, A:F42, A:C43, A:P44, A:R45, A:R46, A:Y47, A:Q49, A:I50, A:G51, A:T52, A:C53, A:G54, A:L55, A:P56, A:G57, A:K59, A:C60, A:C61, A:K62, A:K63, A:P64, A:E65, A:A66, A:K69, A:Q70, A:T71 | 32 | 0.647 |
| 4 | A:G97, A:L98, A:G99, A:Y102, A:A103, A:V106 | 6 | 0.606 |
| | **Nucleoprotein** | | |
| 1 | A:M1, A:R2, A:V3, A:L4, A:Y5, A:L6, A:L7, A:F8, A:S9, A:F10 | 10 | 0.908 |
| 2 | A:G23, A:G24, A:I25, A:G26, A:D27, A:T30, A:K33, A:S34, A:G35, A:A36, A:I37, A:H39, A:V41, A:F42, A:C43, A:P44, A:R46, A:Y47, A:K48, A:Q49, A:I50, A:G51, A:T52, A:G57, A:T58, A:K59, A:C60, A:C61, A:K62, A:K63, A:P64, A:E65, A:A66, A:A67, A:A68, A:K69, A:L70, A:K71, A:E72, A:K73, A:S74, A:S75, A:L76, A:R77, A:G129, A:I130, A:Q131, A:L132, A:D133, A:Q134, A:S146, A:I149, A:Q150, A:L151, A:D152, A:Q153, A:K154, A:P164, A:G170, A:I171, A:Q172, A:L173, A:D174, A:Q175, A:K176, A:D195, A:P196, A:D197, A:D198, A:V199, A:N200, A:K201, A:S202, A:T203, A:L204, A:Q205, A:K206, A:K207, A:F208, A:P209, A:A210, A:Q211, A:V212, A:K213, A:A214, A:R215, A:N216, A:I217, A:I218, A:S219, A:P220, A:V221, A:M222, A:G223, A:V224, A:I225, A:G226, A:F227 | 98 | 0.66 |
| 3 | A:G185, A:K186, A:D187, A:K190 | 4 | 0.6 |
| 4 | A:L11, A:F14, A:L15, A:P17, A:L18, A:V21, A:F22, A:Y93, A:K94, A:L95 | 10 | 0.59 |
| 5 | A:M119, A:L120, A:G121, A:P122, A:G123, A:P124, A:G125, A:Q126, A:S127, A:M128, A:G141, A:P142, A:G143, A:P144, A:G145, A:M160 | 16 | 0.513 |
| | **Proteome-wide Vaccine** | | |
| 1 | A:A91, A:A92, A:Y93, A:H94, A:T95, A:V96, A:G97, A:L98, A:G99, A:Q100, A:G101, A:Y102, A:A103, A:Y105 | 14 | 0.815 |
| 2 | A:P260, A:G261, A:L262, A:L265, A:L268, A:R269, A:V270, A:L271, A:T272, A:F273, A:S274, A:C275, A:S276, A:H277, A:Y278, A:T279, A:N280, A:E281, A:K282, A:K283, A:K284, A:T285, A:D286, A:L287, A:E288, A:L289, A:L293, A:P294, A:S295, A:S296, A:S297, A:Y299, A:S300, A:Y301, A:R303, A:N320, A:K321, A:S322, A:Q325, A:K326, A:F328, A:P329, A:A330, A:Q331, A:V332, A:K333, A:A334, A:R335, A:N336, A:I337, A:I338, A:S339, A:P340, A:V341, A:M342, A:G343, A:V344, A:I345, A:G346, A:F347 | 60 | 0.741 |
| 3 | A:R2, A:V3, A:Y138, A:S156, A:G157, A:P158, A:G159, A:P160, A:G161, A:L162, A:V163, A:V164, A:I165, A:L166, A:I167, A:L168, A:S169, A:I170, A:I171, A:M172, A:F173, A:S174, A:V175, A:L176, A:G177, A:P178, A:G179, A:P180, A:G181, A:V182, A:V184, A:V185, A:I186, A:I188, A:L189, A:S190, A:I191, A:I192, A:M193, A:F194, A:S195, A:V196, A:G197, A:P198, A:G199, A:P200, A:G201, A:M202 | 48 | 0.737 |
| 4 | A:V21, A:F22, A:A127, A:A128, A:Y129, A:K130, A:L131, A:K132, A:K133, A:K134, A:S135, A:A136, A:F137 | 13 | 0.686 |
| 5 | A:G23, A:G24, A:I25, A:G26, A:D27, A:T30, A:S34, A:G35, A:A36, A:I37, A:C38, A:H39, A:P40, A:V41, A:F42, A:C43, A:P44, A:R46, A:Y47, A:K48, A:Q49, A:I50, A:G51, A:T52, A:C53, A:G54, A:L55, A:P56, A:G57, A:T58, A:K59, A:C60, A:C61, A:K62, A:K63, A:E65, A:A66, A:A67, A:K69, A:Q70 | 40 | 0.653 |

## Immune response simulation analysis

Vaccines are designed to stimulate the immune system without causing the disease, but different vaccines can produce distinct immune responses. To assess the efficacy of our vaccines in enhancing the immune response, we conducted an analysis using simulations that measured

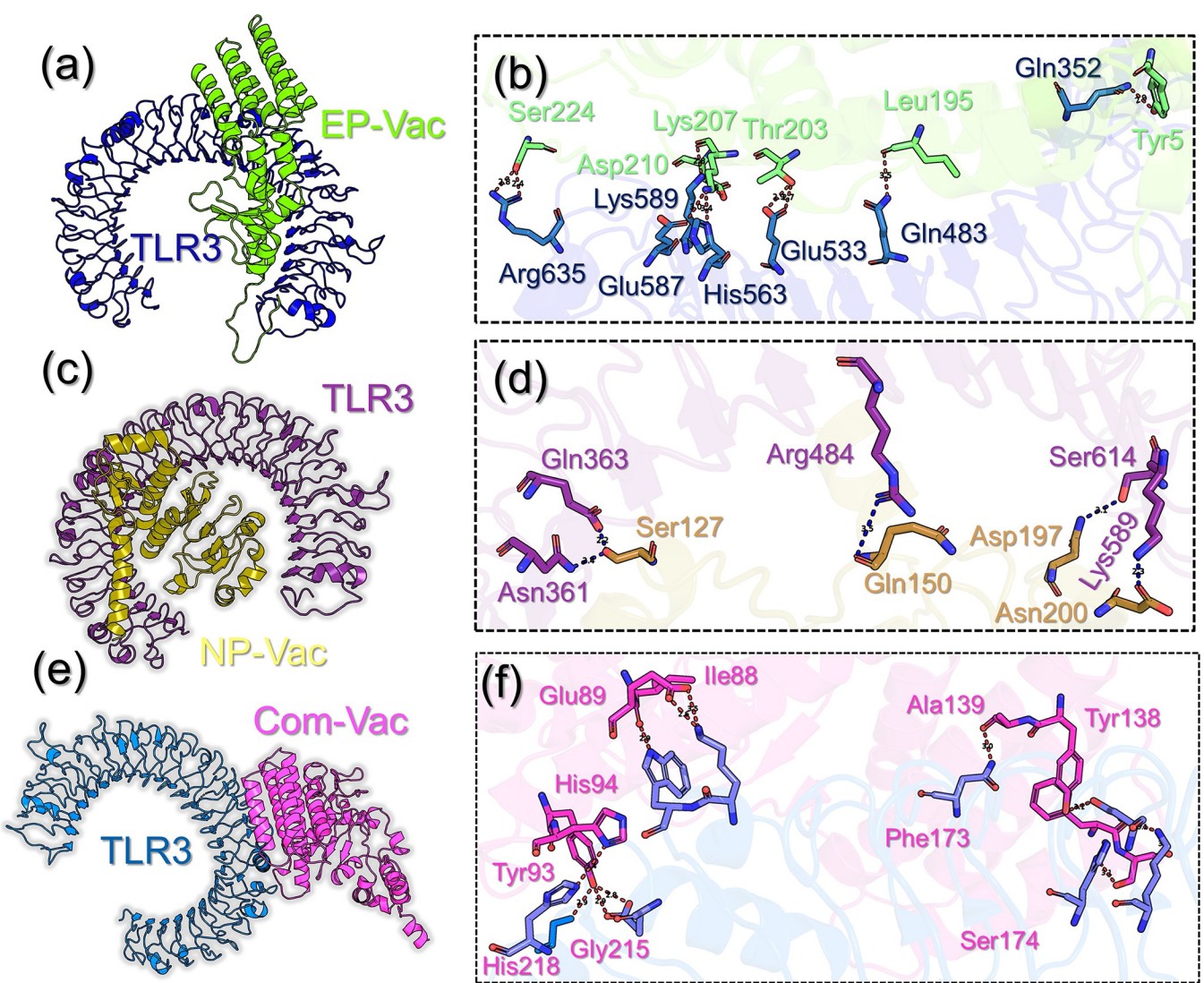

**Fig 8. Interaction pattern of the designed vaccine constructs and TLR3. (a-b)** show the binding mode of Env-Vac with TLR3, **(c-d)** show the interaction pattern of NP-Vac-TLR3 and **(e-f)** show the interaction pattern of com-Vac with TLR3.

antibody production following vaccine administration. The results showed a significant increase in the levels of IgG, IgM, interleukins, and cytokines, as well as antigen clearance over time, after receiving the vaccine and two booster doses. Furthermore, the booster doses led to a further amplification of the immune response, particularly in terms of the combined levels of IgG and IgM, IgG1 and IgG2, and separate levels of IgM and IgG1. The third booster dose resulted in a combined IgG and IgM antibody titer of 670,000–680,000/ml for all vaccines.

**Table 6. Summary of binding free energies for docked complexes (Multi-Epitopes Vaccine constructs-TLR3).**

| Complexes Names | VDW | ELE | GB | SA | TOTAL |
|---|---|---|---|---|---|
| **Env-Vac-TLR3** | -168.66 | -924.98 | 1064.64 | -21.01 | -50.02 |
| **NP-Vac-TLR3** | -221.99 | -1241.91 | 1469.99 | -30.23 | -24.13 |
| **Com-Vac-TLR3** | -149.75 | -911.38 | 1017.53 | -18.69 | -62.30 |

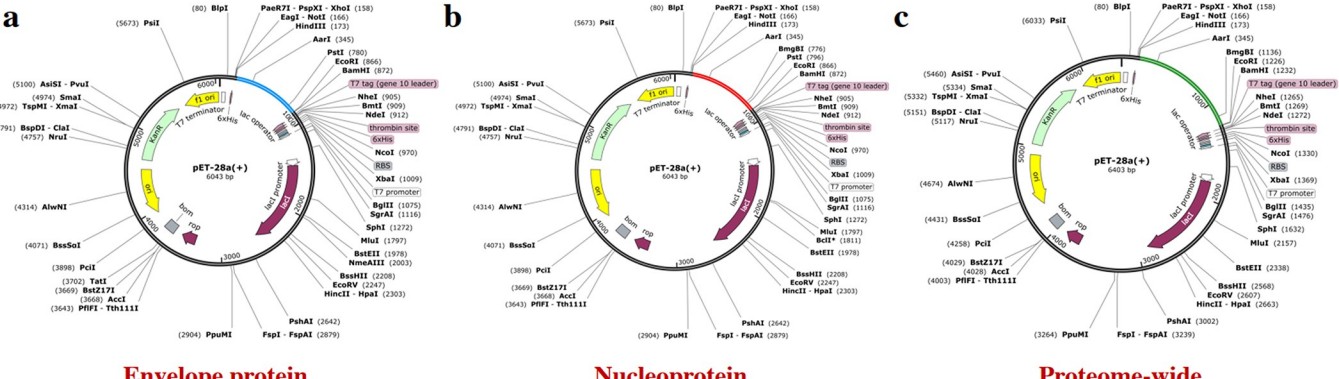

**Fig 9. Cloning the optimized vaccine sequences into the pET28a (+) expression system. (a)** envelope vaccine sequence was cloned into the pET28a (+) vector, **(b)** nucleoprotein vaccine sequence was cloned into the pET28a (+) vector, **(c)** protein-wide vaccine sequence was cloned into the pET28a (+) vector.

Envelope and proteome-wide vaccines had an IgM titer of approximately 380,000/ml, while nucleoprotein vaccines had an IgM titer of around 450,000/ml. Additionally, the level of combined IgG1 and IgG2 was 320,000/ml, 240,000/ml, and 300,000/ml for envelope, nucleoprotein, and proteome-wide vaccines, respectively. Importantly, the level of IgG1 was significantly elevated in response to all vaccine booster injections (**Fig 10A–10C**). These findings indicate that our vaccines effectively stimulate the immune system, generating a robust immune response that can protect against Hantavirus. Following the administration of our vaccines, there was a noticeable increase in the levels of interleukins (IL) and cytokines. Specifically, INF-g levels gradually rose to around 435,000/ml for the designed vaccines, while IL-2 levels

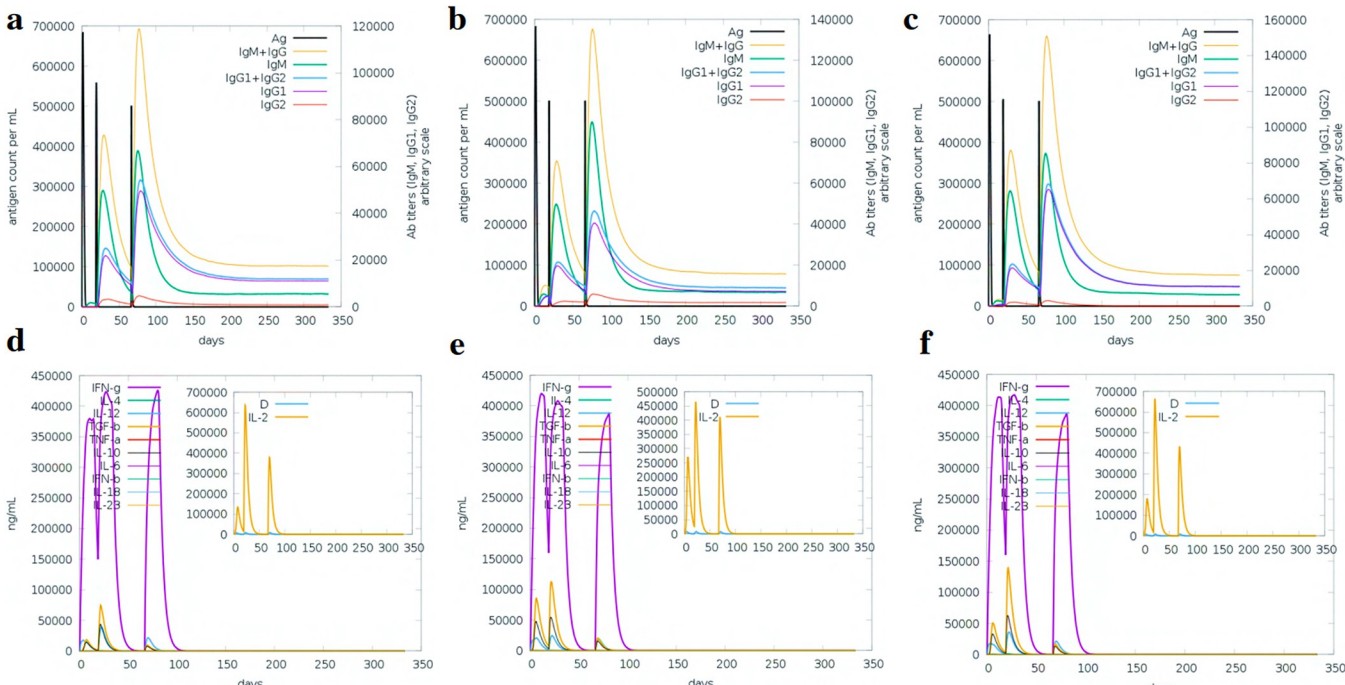

**Fig 10. Immune simulation of the designed vaccine against the Hantavirus. (a-c)** show the predicted antibodies response after three doses while **(d-f)** show the secondary immune elements response upon the injection.

were higher, reaching about 650,000/ml for both envelope and protein-wide vaccines and 450,000/ml for the nucleoprotein vaccines. Additionally, booster doses also resulted in elevated levels of cytokines, including IFN-g, IL-2, IL-10, IL-12, and TGF-b. Our vaccines against Hantavirus were found to be highly immunogenic, inducing a robust immune response (**Fig 10D–10F**).

## Discussion

Developing a multi-epitope vaccine against Hantaviruses, a leading cause of respiratory diseases is a significant challenge in computational drug design. While antibiotics have traditionally been used to treat infectious diseases, their declining effectiveness has prompted researchers to explore alternative approaches like vaccines. Initially developed to control infectious diseases, vaccines have proven highly successful, prompting efforts to develop more affordable, safe, and efficient vaccine production methods. To address the limitations of traditional vaccine technology, such as the risk of virulence recurrence and low cost-effectiveness, a new approach called immunoinformatics has emerged [14, 38–40]. Immunoinformatics leverages an in-depth understanding of pathogen genomes and proteomics to target the immune system using specific epitopes (antigenic regions) from the infectious agent [41]. This approach utilizes databases, web servers, and computational programs, offering a cost-effective, high-quality, safe, and potentially more efficacious method for vaccine development. Numerous vaccines targeting human infections have been created using epitope prediction techniques, including a potential vaccine candidate against the parasite Plasmodium vivax based on the AMA-1 epitope [42]. In summary, the development of a multi-epitope vaccine against hantaviruses through immunoinformatics is a priority due to the pathogen's significant health impact. This computational approach aims to overcome the limitations of traditional vaccine development by leveraging genomic and proteomic data to design epitope-based vaccines in a more efficient, cost-effective, and potentially safer manner.

Essential and antigenic proteins that are also non-allergenic act as primary target for the designing of highly protective and novel vaccine candidates against the infectious diseases. Hence, we collected the whole proteome of hantaviruses containing Envelope polyprotein, Replicase, and Nucleoprotein. The proteome was systematically analyzed to select the highly antigenic and immunological proteins that were processed for the identification of CTL, HTL and B cell epitopes. This is a common approach in vaccine designing where highly antigenic epitopes are mapped and joined together through suitable linkers and the antigenicity is augmented by adding different adjuvants [43]. Moreover, prior to structural modeling the physicochemical properties of the developed vaccines candidates were evaluated which revealed that the designed vaccine is experimentally feasible and are according the previous research publish elsewhere [44]. Additionally, 3D structures were constructed and the proper folding and secondary structure ensured the functionality of the vaccines. The findings align with the previous literature where specific information on the best structural folding and secondary structural elements distribution are discussed in detail [45]. Further refinements were made using Chimera software. Next, a more thorough examination of the interaction with innate immune cells' TLRs was conducted through HDOCK and HawkDock docking analysis. Efficient expression was a priority, and since the JCat server automates reverse transcription, codon adaptation index (CAI) calculations, and GC content calculations for codon optimization, it was chosen for these tasks. The expression vector was Escherichia coli, and to enhance vaccine stability during *E. coli* expression, a disulfide bond was incorporated into the vaccine design. In the current study, immunoinformatics techniques were employed to produce a stable, highly efficacious, and highly expressed vaccine in the *E. coli* translation vector. Clinical trials are necessary to verify its efficacy in humans. The developed MEVC constructs can be

employed in clinical trials to assess the vaccine candidate's potency, safety, and efficacy both *in vitro* and *in vivo*.

## Conclusions

Computational vaccines designing is currently highly accurate, time and cost-effective approach for designing effective vaccines against different diseases. In the current study we shortlisted highly antigenic proteins i.e., envelope, nucleoprotein was from the proteome of hantavirus and subjected to the selection of highly antigenic epitopes to design of next-generation multi-epitope vaccine constructs. We designed Env-Vac, NP-Vac, and Com-Vac constructs, which exhibit stronger antigenic, non-allergenic and favorable physiochemical properties. Moreover, the 3D-structures were predicted and docking, total free energy, *in silico* cloning and immune simulation results for Env-Vac, NP-Vac, and Com-Vac constructs revealed that these designs are highly immunogenic, inducing a robust immune response. The study relies heavily on in silico (computational) methods, which need to be complemented with experimental validation to confirm the efficacy and safety of the identified targets or vaccine constructs. Moreover, there is potential for variability in human immune responses that may not be fully captured by the computational models, given the complexity of the immune system. The long-term stability and immunogenicity of the designed vaccine constructs require further evaluation through clinical trials to assess their performance in human subjects over time.

## Supporting information

**S1 File. Table S1-S5: T cell epitopes, HTL epitopes, B cell epitopes, physiochemical properties, binding free energy.**
(PDF)

## Acknowledgments

This work was supported by Qatar National Research Fund [grant No. NPRP14S-0406-210150] and Qatar University grant No. QUPD-CPH-23/24-592. The statements made herein are solely the responsibility of the authors. Open Access funding is provided by the Qatar National Library. Thank you to the authors for the collaborative work and their cooperation. We would extend our thanks to National University of Medical Sciences for providing us a platform to facilitate present study.

## Author Contributions

**Conceptualization:** Liaqat Ali, Abbas Khan, Taimoor Khan, Yasir Waheed.

**Data curation:** Sobiah Rauf, Abbas Khan, Fahad M. Alshabrmi, Taimoor Khan, Muhammad Suleman, Yasir Waheed, Abdelali Agouni.

**Formal analysis:** Liaqat Ali, Abbas Khan, Rabail Zehra Raza, Taimoor Khan, Muhammad Suleman, Yasir Waheed.

**Funding acquisition:** Abbas Khan, Yasir Waheed.

**Investigation:** Sobiah Rauf, Samreen Rasool, Fahad M. Alshabrmi, Muhammad Suleman.

**Methodology:** Samreen Rasool, Taimoor Khan, Muhammad Suleman, Yasir Waheed, Anwar Mohammad.

**Project administration:** Sobiah Rauf, Taimoor Khan, Yasir Waheed.

**Resources:** Rabail Zehra Raza, Fahad M. Alshabrmi, Muhammad Suleman.

**Software:** Rabail Zehra Raza, Yasir Waheed, Abdelali Agouni.

**Supervision:** Sobiah Rauf, Abbas Khan.

**Validation:** Samreen Rasool, Rabail Zehra Raza.

**Visualization:** Liaqat Ali, Samreen Rasool, Rabail Zehra Raza, Fahad M. Alshabrmi, Abdelali Agouni.

**Writing – original draft:** Liaqat Ali, Abbas Khan, Abdelali Agouni.

**Writing – review & editing:** Abbas Khan, Samreen Rasool, Anwar Mohammad, Abdelali Agouni.

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
