## [Decision Letter · Decision Letter 0]

29 Feb 2024

PONE-D-23-39109Computational Immune Epitopes Screening in the Proteome of Hantavirus to Design next generation Highly Antigenic Vaccine and Validation using Immune SimulationPLOS ONE

Dear Dr. Khan,

Thank you for submitting your manuscript to PLOS ONE. After careful consideration, we feel that it has merit but does not fully meet PLOS ONE’s publication criteria as it currently stands. Therefore, we invite you to submit a revised version of the manuscript that addresses the points raised during the review process.

We look forward to receiving your revised manuscript.

Kind regards,

Sheikh Arslan Sehgal, PhD

Academic Editor

PLOS ONE

4. We suggest you thoroughly copyedit your manuscript for language usage, spelling, and grammar. If you do not know anyone who can help you do this, you may wish to consider employing a professional scientific editing service.  

"N/A"

6. In the online submission form, you indicated that data will be furnished upon a reasonable request.

Reviewers' comments:

Reviewer's Responses to Questions

**Comments to the Author**

1. Is the manuscript technically sound, and do the data support the conclusions?

Reviewer #1: Yes

Reviewer #2: Yes

2. Has the statistical analysis been performed appropriately and rigorously? 

Reviewer #1: N/A

Reviewer #2: Yes

3. Have the authors made all data underlying the findings in their manuscript fully available?

Reviewer #1: Yes

Reviewer #2: Yes

4. Is the manuscript presented in an intelligible fashion and written in standard English?

Reviewer #1: Yes

Reviewer #2: Yes

5. Review Comments to the Author

Reviewer #1: The manuscript could make a nice contribution to the field; however, it requires a major revision before it is suitable for publication.

1. The title of the manuscript is an important factor in attracting the audience, it is better to use a more attractive title for the manuscript.

2. The inappropriate use of the English language in terms of grammar and writing style is deficient. The whole manuscript must be checked by an expert language editing service or a native speaker with excellent writing skills.

3. In line 115, the phrase "antigenic linkers" is meaningless, please use only the word "linkers".

4. Solubility of the vaccine construct is an important feature. Authors should evaluate the solubility of the vaccine construct by existing servers.

5. Authors should predict discontinuous B cell epitopes on the 3D structure of the vaccine construct.

6. The authors should perform disulfide engineering of the 3D structure of the structure.

7. The discussion section is one of the important sections of the article that this article lacks and in fact the article is structurally incomplete, the authors should add the discussion section to the article and compare the results of their work with the results of similar works, for this purpose the authors They should use and cite the following articles as references.

https://doi.org/10.1007/s12033-023-00949-y

https://doi.org/10.1080/07391102.2023.2258403

https://doi.org/10.2174/1573409919666230612125440

https://doi.org/10.1371%2Fjournal.pone.0286224

https://doi.org/10.3390%2Fvaccines11020263

https://doi.org/10.1038/s41598-022-11851-z

Reviewer #2: The manuscript by Khan et al uses immunoinformatics workflow to design a vaccine for Hantaviruses. The work is systematically sound and publishable however minor revision is recommended. The specific comments are;

1. Add the limitation of the current study in the conclusion.

2. In Line 70-73, a proper reference should be added.

3. “Toll-like receptor (TLR3)” is already defined, use abbreviation only in the later part.

4. Check the link http://www.ddg-pharmfac.net/vaxijen/VaxiJen/VaxiJen.html

5. In Line 143 “[25]” reference is not valid.

6. Escherichia coli (E. coli) should be italic.

7. Add reference for “Immune simulation”.

6. PLOS authors have the option to publish the peer review history of their article (what does this mean?). If published, this will include your full peer review and any attached files.

Reviewer #1: No

Reviewer #2: **Yes: **Sobia Ahsan Halim

---

## [Author Response · Author response to Decision Letter 0]

22 May 2024

Response to Reviewers

Dear Editor,

We would like to submit our revised manuscript entitled “In Silico Design of Multi-Epitope Vaccines against the Hantaviruses by Integrated Structural Vaccinology and Molecular Modeling Approaches” to your journal for possible publication. We have addressed all the issues raised by the reviewers and we thank them for the thorough reading of the manuscript that has significantly improved the manuscript. 

Yours,

Dr. Abdelali Agouni,

Qatar University, Doha, Qatar.

aagouni@qu.edu.qa

Reviewer #1: The manuscript could make a nice contribution to the field; however, it requires a major revision before it is suitable for publication.

1. The title of the manuscript is an important factor in attracting the audience, it is better to use a more attractive title for the manuscript.

Response: Thank you very much for this important point. we have now changed the title as “In Silico Design of Multi-Epitope Vaccines against the Hantaviruses by Integrated Structural Vaccinology and Molecular Modeling Approaches”.

2. The inappropriate use of the English language in terms of grammar and writing style is deficient. The whole manuscript must be checked by an expert language editing service or a native speaker with excellent writing skills.

Response: Thank you very much for this important point. the whole manuscript is now edited and rectified for language mistakes. 

3. In line 115, the phrase "antigenic linkers" is meaningless, please use only the word "linkers".

Response: Thank you very much for this important point. We have now corrected it as suggested.

4. Solubility of the vaccine construct is an important feature. Authors should evaluate the solubility of the vaccine construct by existing servers.

Response: Thank you very much for this important point. the solubility is now calculated. 

“The solubility for each construct was estimated by using online server Protein-sol (https://protein-sol.manchester.ac.uk/ ) which calculate the solubility of a protein by comparing with the experimental dataset [3].” “Moreover, proteins with a scaled solubility value above 0.45 are predicted to be more soluble than the average soluble Escherichia coli protein from the experimental solubility dataset. For our proteins, i.e., for the Env-Vac the predicted solubility was 0.628, for the NP-Vac the predicted solubility was 0.641 while for the Com-Vac the solubility was calculated to 0.544. The solubility assessment revealed that all the designed vaccine candidates are highly soluble in the ensures that the vaccine components can be properly dissolved and administered within the body. The solubility graphs are given in Figures 6a-6c.” 

Figure 6: Solubility analysis of each vaccine construct. (A) show the solubility graph for Env-Vac, (B) show the solubility graph for NP-Vac while (C) show the solubility graph for Com-Vac. 

5. Authors should predict discontinuous B cell epitopes on the 3D structure of the vaccine construct.

Response: Thank you very much for this important point. the conformational B cell epitopes are now predicted and added to the results.

Conformational B cell epitopes prediction

For the conformational B cell epitopes in each vaccine construct we used Ellipro server. For the Env-Vac four conformational B cell epitopes were reported while for the NP-Vac and Com-Vac five conformational B cell epitopes in each were calculated. The CBCEs in Env-Vac are given in Figures 7a-7d, the CBCEs in NP-Vac are given in Figures 7e-7i while the CBCEs in Com-Vac are given in Figures 7j-7n. The specific residues, scores and other information are given in Table 5.

Figure 7: The predicted conformational B cell epitopes for each vaccine construct. (a-d) shows the CBCEs in Env-Vac, (e-i) shows the CBCEs in NP-Vac while (j-n) show the CBCEs in Com-Vac

Table 5: The predicted B cell conformational epitopes with their residues, size and scores.

S. No Residues Size Score

Envelope Protein

1 A:T203, A:E205, A:K206, A:K207, A:K208, A:T209, A:D210, A:L211, A:E212, A:L213, A:D214, A:F215, A:S216, A:L217, A:P218, A:S219, A:S220, A:S221, A:S222, A:Y223, A:S224, A:Y225, A:R227 23 0.893

2 A:M1, A:R2, A:V3, A:L4, A:Y5, A:L6, A:L7, A:F8, A:V110, A:I113, A:L114, A:I116, A:I117, A:M118, A:F119, A:S120, A:G121, A:P122, A:G123, A:P124, A:G125, A:V126, A:V127, A:V128, A:V130, A:V131, A:I134, A:L135, A:I137, A:I138, A:M139, A:G141, A:P142, A:I154, A:I155, A:M156, A:F157, A:S158, A:V159, A:L160, A:G161, A:P162, A:G163, A:P164, A:G165, A:V166, A:V168 47 0.722

3 A:G24, A:I25, A:G26, A:S34, A:G35, A:V41, A:F42, A:C43, A:P44, A:R45, A:R46, A:Y47, A:Q49, A:I50, A:G51, A:T52, A:C53, A:G54, A:L55, A:P56, A:G57, A:K59, A:C60, A:C61, A:K62, A:K63, A:P64, A:E65, A:A66, A:K69, A:Q70, A:T71 32 0.647

4 A:G97, A:L98, A:G99, A:Y102, A:A103, A:V106 6 0.606

Nucleoprotein

1 A:M1, A:R2, A:V3, A:L4, A:Y5, A:L6, A:L7, A:F8, A:S9, A:F10 10 0.908

2 A:G23, A:G24, A:I25, A:G26, A:D27, A:T30, A:K33, A:S34, A:G35, A:A36, A:I37, A:H39, A:V41, A:F42, A:C43, A:P44, A:R46, A:Y47, A:K48, A:Q49, A:I50, A:G51, A:T52, A:G57, A:T58, A:K59, A:C60, A:C61, A:K62, A:K63, A:P64, A:E65, A:A66, A:A67, A:A68, A:K69, A:L70, A:K71, A:E72, A:K73, A:S74, A:S75, A:L76, A:R77, A:G129, A:I130, A:Q131, A:L132, A:D133, A:Q134, A:S146, A:I149, A:Q150, A:L151, A:D152, A:Q153, A:K154, A:P164, A:G170, A:I171, A:Q172, A:L173, A:D174, A:Q175, A:K176, A:D195, A:P196, A:D197, A:D198, A:V199, A:N200, A:K201, A:S202, A:T203, A:L204, A:Q205, A:K206, A:K207, A:F208, A:P209, A:A210, A:Q211, A:V212, A:K213, A:A214, A:R215, A:N216, A:I217, A:I218, A:S219, A:P220, A:V221, A:M222, A:G223, A:V224, A:I225, A:G226, A:F227 98 0.66

3 A:G185, A:K186, A:D187, A:K190 4 0.6

4 A:L11, A:F14, A:L15, A:P17, A:L18, A:V21, A:F22, A:Y93, A:K94, A:L95 10 0.59

5 A:M119, A:L120, A:G121, A:P122, A:G123, A:P124, A:G125, A:Q126, A:S127, A:M128, A:G141, A:P142, A:G143, A:P144, A:G145, A:M160 16 0.513

 Proteome-wide Vaccine 

1 A:A91, A:A92, A:Y93, A:H94, A:T95, A:V96, A:G97, A:L98, A:G99, A:Q100, A:G101, A:Y102, A:A103, A:Y105 14 0.815

2 A:P260, A:G261, A:L262, A:L265, A:L268, A:R269, A:V270, A:L271, A:T272, A:F273, A:S274, A:C275, A:S276, A:H277, A:Y278, A:T279, A:N280, A:E281, A:K282, A:K283, A:K284, A:T285, A:D286, A:L287, A:E288, A:L289, A:L293, A:P294, A:S295, A:S296, A:S297, A:Y299, A:S300, A:Y301, A:R303, A:N320, A:K321, A:S322, A:Q325, A:K326, A:F328, A:P329, A:A330, A:Q331, A:V332, A:K333, A:A334, A:R335, A:N336, A:I337, A:I338, A:S339, A:P340, A:V341, A:M342, A:G343, A:V344, A:I345, A:G346, A:F347 60 0.741

3 A:R2, A:V3, A:Y138, A:S156, A:G157, A:P158, A:G159, A:P160, A:G161, A:L162, A:V163, A:V164, A:I165, A:L166, A:I167, A:L168, A:S169, A:I170, A:I171, A:M172, A:F173, A:S174, A:V175, A:L176, A:G177, A:P178, A:G179, A:P180, A:G181, A:V182, A:V184, A:V185, A:I186, A:I188, A:L189, A:S190, A:I191, A:I192, A:M193, A:F194, A:S195, A:V196, A:G197, A:P198, A:G199, A:P200, A:G201, A:M202 48 0.737

4 A:V21, A:F22, A:A127, A:A128, A:Y129, A:K130, A:L131, A:K132, A:K133, A:K134, A:S135, A:A136, A:F137 13 0.686

5 A:G23, A:G24, A:I25, A:G26, A:D27, A:T30, A:S34, A:G35, A:A36, A:I37, A:C38, A:H39, A:P40, A:V41, A:F42, A:C43, A:P44, A:R46, A:Y47, A:K48, A:Q49, A:I50, A:G51, A:T52, A:C53, A:G54, A:L55, A:P56, A:G57, A:T58, A:K59, A:C60, A:C61, A:K62, A:K63, A:E65, A:A66, A:A67, A:K69, A:Q70 40 0.653

6. The authors should perform disulfide engineering of the 3D structure of the structure.

Response: Thank you very much for this important point. NO specific disulfide bond was observed in each construct. 

7. The discussion section is one of the important sections of the article that this article lacks and in fact the article is structurally incomplete, the authors should add the discussion section to the article and compare the results of their work with the results of similar works, for this purpose the authors They should use and cite the following articles as references.

https://doi.org/10.1007/s12033-023-00949-y

https://doi.org/10.1080/07391102.2023.2258403

https://doi.org/10.2174/1573409919666230612125440

https://doi.org/10.1371%2Fjournal.pone.0286224

https://doi.org/10.3390%2Fvaccines11020263

https://doi.org/10.1038/s41598-022-11851-z

Response: Thank you very much for this important point. we have now added more discussion by citing the aforementioned literature.

“Developing a multi-epitope vaccine against Hantaviruses, a leading cause of respiratory diseases is a significant challenge in computational drug design. While antibiotics have traditionally been used to treat infectious diseases, their declining effectiveness has prompted researchers to explore alternative approaches like vaccines. Initially developed to control infectious diseases, vaccines have proven highly successful, prompting efforts to develop more affordable, safe, and efficient vaccine production methods. To address the limitations of traditional vaccine technology, such as the risk of virulence recurrence and low cost-effectiveness, a new approach called immunoinformatics has emerged [14, 38-40]. Immunoinformatics leverages an in-depth understanding of pathogen genomes and proteomics to target the immune system using specific epitopes (antigenic regions) from the infectious agent [41]. This approach utilizes databases, web servers, and computational programs, offering a cost-effective, high-quality, safe, and potentially more efficacious method for vaccine development. Numerous vaccines targeting human infections have been created using epitope prediction techniques, including a potential vaccine candidate against the parasite Plasmodium vivax based on the AMA-1 epitope [42]. In summary, the development of a multi-epitope vaccine against hantaviruses through immunoinformatics is a priority due to the pathogen's significant health impact. This computational approach aims to overcome the limitations of traditional vaccine development by leveraging genomic and proteomic data to design epitope-based vaccines in a more efficient, cost-effective, and potentially safer manner.

Essential and antigenic proteins that are also non-allergenic act as primary target for the designing of highly protective and novel vaccine candidates against the infectious diseases. Hence, we collected the whole proteome of hantaviruses containing Envelope polyprotein, Replicase, and Nucleoprotein. The proteome was systematically analyzed to select the highly antigenic and immunological proteins that were processed for the identification of CTL, HTL and B cell epitopes. This is a common approach in vaccine designing where highly antigenic epitopes are mapped and joined together through suitable linkers and the antigenicity is augmented by adding different adjuvants [43]. Moreover, prior to structural modeling the physicochemical properties of the developed vaccines candidates were evaluated which revealed that the designed vaccine is experimentally feasible and are according the previous research publish elsewhere [44]. Additionally, 3D structures were constructed and the proper folding and secondary structure ensured the functionality of the vaccines. The findings align with the previous literature where specific information on the best structural folding and secondary structural elements distribution are discussed in detail [45]. Further refinements were made using Chimera software. Next, a more thorough examination of the interaction with innate immune cells' TLRs was conducted through HDOCK and HawkDock docking analysis. Efficient expression was a priority, and since the JCat server automates reverse transcription, codon adaptation index (CAI) calculations, and GC content calculations for codon optimization, it was chosen for these tasks. The expression vector was Escherichia coli, and to enhance vaccine stability during E. coli expression, a disulfide bond was incorporated into the vaccine design. In the current study, immunoinformatics techniques were employed to produce a stable, highly efficacious, and highly expressed vaccine in the E. coli translation vector. Clinical trials are necessary to verify its efficacy in humans. The developed MEVC constructs can be employed in clinical trials to assess the vaccine candidate's potency, safety, and efficacy both in vitro and in vivo.”

Reviewer #2: The manuscript by Khan et al uses immunoinformatics workflow to design a vaccine for Hantaviruses. The work is systematically sound and publishable however minor revision is recommended. The specific comments are;

1. Add the limitation of the current study in the conclusion.

Response: Thank you very much for this important point. the limitations are now added in the conclusion section.

“The study relies heavily on in silico (computational) methods, which need to be complemented with experimental validation to confirm the efficacy and safety of the identified targets or vaccine constructs. Moreover, there is potential for variability in human immune responses that may not be fully captured by the computational models, given the complexity of the immune system. The long-term stability and immunogenicity of the designed vaccine constructs require further evaluation through clinical trials to assess their performance in human subjects over time.”

2. In Line 70-73, a proper reference should be added.

Response: Thank you very much for this important point. a proper reference has now been added. 

3. “Toll-like receptor (TLR3)” is already defined, use abbreviation only in the later part.

Response: Thank you very much for this important point. checked and corrected now.

4. Check the link http://www.ddg-pharmfac.net/vaxijen/VaxiJen/VaxiJen.html

Response: Thank you very much for this important point. the link is now checked and working properly.

5. In Line 143 “[25]” reference is not valid.

Response: Thank you very much for this important point. a proper reference is now added to the server location. 

6. Escherichia coli (E. coli) should be italic.

Response: Thank you very much for this important point. changed the format as required. 

7. Add reference for “Immune simulation”.

Response: Thank you very much for this important point. A proper reference is now added to the server location.

---

## [Decision Letter · Decision Letter 1]

30 May 2024

In Silico Design of Multi-Epitope Vaccines against the Hantaviruses by Integrated Structural Vaccinology and Molecular Modeling Approaches

PONE-D-23-39109R1

Dear Dr. Khan,

We’re pleased to inform you that your manuscript has been judged scientifically suitable for publication and will be formally accepted for publication once it meets all outstanding technical requirements.

Kind regards,

Sheikh Arslan Sehgal, PhD

Academic Editor

PLOS ONE

Additional Editor Comments (optional):

Reviewers' comments:

Reviewer's Responses to Questions

**Comments to the Author**

1. If the authors have adequately addressed your comments raised in a previous round of review and you feel that this manuscript is now acceptable for publication, you may indicate that here to bypass the “Comments to the Author” section, enter your conflict of interest statement in the “Confidential to Editor” section, and submit your "Accept" recommendation.

Reviewer #1: All comments have been addressed

Reviewer #2: All comments have been addressed

2. Is the manuscript technically sound, and do the data support the conclusions?

Reviewer #1: Yes

Reviewer #2: Yes

3. Has the statistical analysis been performed appropriately and rigorously? 

Reviewer #1: N/A

Reviewer #2: N/A

4. Have the authors made all data underlying the findings in their manuscript fully available?

Reviewer #1: Yes

Reviewer #2: Yes

5. Is the manuscript presented in an intelligible fashion and written in standard English?

Reviewer #1: Yes

Reviewer #2: Yes

6. Review Comments to the Author

Reviewer #1: I thank the authors for carefully considering all the comments and this article deserves to be a model for authors in the field of vaccine design.

Reviewer #2: The authors have addressed all the comments according to the suggested corrections, and the revised manuscript can be accepted.

7. PLOS authors have the option to publish the peer review history of their article (what does this mean?). If published, this will include your full peer review and any attached files.

Reviewer #1: No

Reviewer #2: **Yes: **Sobia Ahsan Halim

---

## [Editor Report · Acceptance letter]

12 Jul 2024

PONE-D-23-39109R1 

PLOS ONE

Dear Dr. Khan, 

I'm pleased to inform you that your manuscript has been deemed suitable for publication in PLOS ONE. Congratulations! Your manuscript is now being handed over to our production team.

Kind regards, 

on behalf of

Dr Sheikh Arslan Sehgal 

Academic Editor

PLOS ONE